# Mapping Invasion Potential of the Pest from Central Asia, *Trypophloeus klimeschi* (Coleoptera: Curculionidae: Scolytinae), in the Shelter Forests of Northwest China

**DOI:** 10.3390/insects12030242

**Published:** 2021-03-12

**Authors:** Hang Ning, Ming Tang, Hui Chen

**Affiliations:** 1State Key Laboratory for Conservation and Utilization of Subtropical Agro-Bioresources, South China Agricultural University, Guangzhou 510642, China; ninghang@nwafu.edu.cn (H.N.); tangmingyl@163.com (M.T.); 2College of Forestry, Northwest A&F University, Yangling 712100, China

**Keywords:** *Trypophloeus klimeschi*, climate change, species distribution models, insect–climate interactions, pest management

## Abstract

**Simple Summary:**

*Trypophloeus klimeschi* Eggers (Coleoptera: Curculionidae: Scolytinae) causes substantial mortality to *Populus alba* var. *pyramidalis* individuals in Xinjiang, China. Currently, the number of host trees killed by this bark beetle is increasing. Climate change has exacerbated this problem, causing its range to expand. Here, using Maxent, we simulated its distributions in the 2030s and 2050s under representative concentration pathways (RCPs) 2.6, 4.5 and 8.5. The distribution expanded the most under RCP 8.5. It is large enough to pose substantial challenges for forest managers across northern China. Our study contributes to the construction and protection of shelter forests in Northern China because we present novel evidence of the potential impacts of climate change on forestry.

**Abstract:**

Temperature and precipitation are the two main factors constraining the current distribution of *Trypophloeus klimeschi*. Currently, *T. klimeschi* is mainly distributed in South Xinjiang, where it occurs between the southern edge of the Tianshan Mountains and northern edge of the Tarim Basin. In addition, Dunhuang in northern Gansu also provide suitable habitats for this bark beetle. Two other potential areas for this species are in or near the cities of Alaer and Korla. Under future climate scenarios, its total suitable area is projected to increase markedly over time. Among the climate scenarios, the distribution expanded the most under the maximum greenhouse gas emission scenario (representative concentration pathway (RCP) 8.5). Jiuquan in Gansu is projected to become a suitable area in the 2030s. Subsequently, *T. klimeschi* is expected to enter western Inner Mongolia along the Hexi Corridor in the 2050s. In southeastern Xinjiang, however, the suitable area in northern Ruoqiang and most areas of Korla may decrease. By the 2050s, it is large enough to pose substantial challenges for forest managers across northern China. Our findings provide information that can be used to monitor *T. klimeschi* populations, host health, and the impact of climate change, shedding light on the effectiveness of management responses.

## 1. Introduction

*Trypophloeus klimeschi* Eggers (Coleoptera: Curculionidae: Scolytinae) is a newly recorded species in China [1]. *T. klimeschi* is native to the Kyrgyz Republic, which borders the Xinjiang Uygur Autonomous Region [2]. In 2003, this beetle was first found in Aksu, Xinjiang [1,3]. Subsequently, the beetle spread rapidly to adjacent areas and is now found in Dunhuang, Gansu Province [4]. *T. klimeschi* first attacks the branch shoots of *Populus alba* var. *pyramidalis* Bunge and then gradually spreads to the main trunk. The beetle completes its life cycle in the phloem under the bark, except during a brief dispersal when the adults search for new host trees [4,5]. The injured branches turn yellow and wither, many of the leaves fall off, and dense holes are formed in the trunk surface, causing the injured trees to wither and rapidly die [1,4]. *Trypophloeus klimeschi* typically completes two generations per year and mature larvae overwinter. First-generation adults emerge from late-May to late-June and second-generation adults emerge in August [6]. *T. klimeschi* have strong host-selection specificity, which can cause the extent of injury to vary with the age of *P. alba* var. *pyramidalis*. However, other local poplar species, such as *P. alba* L., *P. tomentosa* Carr, and *P. dakuanensis* Hsu, are not invaded by this beetle [4]. In recent years, a large number of host trees in northwestern China were killed by this beetle, which overburdened the original fragile shelterbelt [4,6]. As a preferred afforestation tree species for greening and shelterbelts, especially in the northern China, many *P. alba* var. *pyramidalis* stands provide an ecological corridor for the spread of *T. klimeschi* from the northwestern to the eastern region of China [7]. Integrated control of this bark beetle has already been placed on the agenda. However, traditional physical and chemical control measures are unable to effectively treat *T. klimeschi* due to its hidden lifestyle. Based on these reasons, the adverse situation is expected to become increasingly serious.

Global warming, which has been acknowledged internationally in recent years, has had global effects, and further impacts are inevitable [8]. Future changes in temperatures might have significant effects on insect growth rates, thus influencing both biological processes (e.g., number of generations per year) and geographical distribution [9,10]. Insects are therefore particularly sensitive to the warming associated with climate change, and may be early indicators of local climate change [11]. Observations of field populations found that the geographical range of *T. klimeschi* have expanded [6]. New occurrence records continue to be published [4,5,6]. The projected climate changes are likely to alter the characteristics of the spread of *T. klimeschi*, and cause shifts in species distributions on an ongoing basis. It is therefore critical to understand how its geographical distribution in China will respond to climate change, to enable appropriate ongoing forest ecosystem management. To address this, we applied species distribution models (SDMs) to predict climate-change-driven habitat shifts for this species.

SDMs have become a major focus of large-scale ecology and biogeography studies. They are widely applied to model plant and insect habitat ranges; assess the effects of global environmental change on species and ecosystems; evaluate the risk of species invasion and spread; and manage and plan the habitats for species, communities, and ecosystems [12,13]. Popular SDMs, including BIOCLIM, GARP, BIOMAPPER, DOMAIN, GAM, GLM, CLIMEX, and Maxent [14,15,16,17,18,19,20,21], use species occurrence records (presence only, or presence/absence), associated with environmental variables, to describe the fundamental niche of a particular species. They then project this niche onto the landscape of interest to reflect the potential distribution area of the species [22]. Maxent is widely used because of its excellent predictive performance. Maxent modeling predictions are typically stable and reliable even when species distribution data are incomplete or when sample sizes are small [23,24,25].

Effective representation of the geographical distribution of the *T. klimeschi* population can provide a significant reference point for forest managers in the face of climate change. In this study, we used Maxent to model the current and future potential distributions of *T. klimeschi* in China. This research has three objectives: (1) identify the dominant environmental variables that describe the distribution range of *T. klimeschi*; (2) predict future potential distributions of *T. klimeschi* under climate change scenarios; and (3) track the changes in the centroids of *T. klimeschi* distributions.

## 2. Materials and Methods

### 2.1. Study Area

The cultivation area of *P. alba* var. *pyramidalis* was used as a proxy study area (Figure 1). *P. alba* var. *pyramidalis* is a fine tree species for farmland shelterbelt, fast-growing and high-yield forest, windbreak and sand-fixation forest, and roadside greening because of its excellent characteristics, with strong wind resistance, drought resistance, and ease of maintenance [7]. All afforestation projects are flourishing with the long-term support and protection of the Chinese government. The host trees are widely distributed in northern China, especially in Xinjiang. To better understand the climatic environment, the distribution profile of the *P. alba* var. *pyramidalis* is as follows (Table 1) [7,26].

### 2.2. Occurrence Collection

The fieldwork was conducted with the assistance of the Department of Forestry Protection from May to October 2018–2019. According to the infected area, the county forestry departments were regarded as the survey units. Along the edge of the Tarim Basin, the fieldwork first started in Hotan and finally reached Dunhuang. The detailed geographical locations of occurrence area are shown in Figure 2. The distance between the sampling sites was >20 km. When the distance between two or more occurrence points was <20 km, we recorded only one occurrence. Due to the cryptic nature of the bark beetle, traps were used to monitor beetle presence. The trap in this study consisted of six identical black plastic funnels aligned vertically over a cup, similar to those commonly used to collect bark beetles (Appendix A). Generally, the attractant was hung in the middle and lower part of trap. Then, each trap (W × H = 18 × 50 cm, YBQ-LD-004, Geruibiyuan Technology Co., Ltd., Beijing, China) was vertically suspended from a manila rope tied between two host trees at the forest edge and positioned more than 1 m from the closest host tree. The lower end of the trap was 1.5 m away from the ground. The traps were spaced at least 30 m apart. The components of the attractant included nonanal, 2–methylbutanal, decanal, 2–hydroxybenzaldehyde, (Z)−3–hexen–1–ol benzoate, methyl, benzoate, methyl salicylate, and geraniol, based on the behavioral responses of *T. klimeschi* to an olfactory stimulants trial [6]. Then, eight compounds were dissolved in solvent hexane. Finally, the attractant was added into a 15-mL slow-release polyethylene vial (Sino-Czech Trading Company, Beijing, China) at a release speed of 200 mg/day. Liquid paraffin was used as a killing agent in the cups. Trapped *T. klimeschi* were collected weekly, counted, and then stored in 70% ethanol. Location name, longitude, latitude, and altitude were documented for all occurrence points. In addition, occurrence data were collected by consulting the relevant literature. Those points without accurate location data were excluded. For example, Kyrgyz Republic had a few records, but lacked coordinates. As a result, there were only seven points obtained through the literature, and others were from the field survey. Finally, we obtained 89 point-based occurrence records (Figure 1).

We gathered data regarding the presence of *P. alba* var. *pyramidalis* via three methods. First, we recorded *P. alba* var. *pyramidalis* in the field from 2016 to 2018. Similarly, raw data were processed as for the *T. klimeschi* data. Second, we obtained *P. alba* var. *pyramidalis* presence data from the GBIF (http://www.gbif.org (accessed on 11 March 2021)) and the Chinese Virtual Herbarium (CVH, http://www.cvh.org.cn/ (accessed on 11 March 2021)). Those records without coordinates and altitudes, including duplicates, were deleted. Third, we consulted the literature for *P. alba* var. *pyramidalis* records using a Chinese academic website (https://www.cnki.net/ (accessed on 11 March 2021)). As a result, we discarded 31 occurrence records with incomplete geographic information. In total, 324 *P. alba* var. *pyramidalis* occurrence records were used to generate its potential distribution via Maxent.

### 2.3. Environmental Variables

Nineteen bioclimatic variables, three terrain variables, and one host variable were used to predict the potential distribution of *T. klimeschi* (Table 2). Nineteen bioclimatic variables, with a spatial resolution of 2.5 arc minutes, were freely downloaded from www.worldclim.org (accessed on 11 March 2021). Terrain variables were obtained from the United States Geological Survey (USGS, http://edcdaac.usgs.gov/gtopo30/gtopo30.html (accessed on 11 March 2021)). We used Arc-GIS v.10.2 (Environmental Systems Research Institute Inc., Redlands, CA, USA) to convert all variable layers into ASCII format for use with Maxent. We accessed the National Fundamental Geographic Information System (http://www.ngcc.cn (accessed on 11 March 2021)) and downloaded the National Map as the analytical base map.

To determine the future distributions of *T. klimeschi* under different climate scenarios, datasets on future climate from the Climate Change, Agriculture and Food Security program (CCAFS, http://ccafs-climate.org/ (accessed on 11 March 2021)) were used. Representative concentration pathways (RCPs) announced in the fifth Intergovernmental Panel on Climate Change report include four greenhouse gas concentration trajectories, representing scenarios in which the total radiative forcing in 2100 has reached 2.6 W/m^^2^^, 4.5 W/m^^2^^, 6 W/m^^2^^, and 8.5 W/m^^2^^ (in excess of 1750 W/m^^2^^) [27]. The RCPs prescribe levels of radiative forcing arising from different atmospheric concentrations of greenhouse gas that lead to different levels of climate change. For example, RCP 2.6 is projected to lead to global mean temperature changes of about 0.9 °C–2.3 °C, and RCP 8.5 to global mean temperature changes of about 3.2 °C–5.4 °C [28]. There has been a 41% increase in total radiative forcing since 1990. If the current trend in greenhouse gas concentrations continues, mean annual global temperature is expected to increase between 1.8 °C and 4 °C during the 21st century [29]. From the perspective of the global environment, all scenarios are likely to occur. We used the BCC-CSM 1.1 climate system model, developed by the Beijing Climate Center, China. After more than ten years of development, its performance and functions have been effectively improved [30]. It has been widely used for climate prediction. In this study, RCP 2.6, RCP 4.5, and RCP 8.5 were selected to model the future distributions of *T. klimeschi* in the 2030s and 2050s.

To eliminate over-fitting of the models due to multicollinearity within environmental variables, variables with higher maximum entropy gain and non-zero regression coefficients according to the least absolute shrinkage and selection (LASSO) regularization method were selected [31,32]. LASSO is a regression analysis method that performs both the selection of variables and regularization, in order to enhance the prediction accuracy and interpretability of the statistical model. The method regularizes model parameters by shrinking the regression coefficients, reducing some to zero [33]. The feature selection phase occurs after shrinkage; in this phase, every non-zero value is selected for use in the model. This improves model accuracy because coefficient shrinkage reduces variance and minimizes bias. LASSO was applied using the R package glmnet (http://www.web.stanford.edu/~hastie/Papers/Glmnet_Vignette.pdf (accessed on 11 March 2021)).

### 2.4. Species Distribution Modeling

We used Maxent v.3.3.1 (http://www.cs.princeton.edu/wschapire/MaxEnt (accessed on 11 March 2021)) to predict the potential distribution of *T. klimeschi*. Ten-fold cross-validation was performed to train and validate the model. The occurrence dataset was randomly divided into 10 equal-sized subsets. Of these, a single subset was retained for model-testing; the remaining subsets were used as training data. For each subset, 90% of the occurrence data were used to train the single model, and the remaining data were used to test the predictive ability of the model. The cross-validation process was repeated 10 times, with each of the 10 subsets used once as validation data. The 10 results were then be averaged to produce a single estimation [34]. All mathematical modeling process were based on the maxnet function in the maxnet R package (https://www.rdocumentation.org/packages/maxnet/versions/0.1.2 (accessed on 11 March 2021)). The area under the receiver operator characteristic (ROC) curve (AUC) was used to evaluate the predictive performance of each Maxent model. When the ROC curve is at 45° in the ROC space, the AUC value is close to 0.5, indicating that the model is a random model with an accuracy of 50%; at AUC > 0.5, the model is more accurate than a random model; at AUC < 0.5, the model is less accurate than a random model. The closer AUC is to 1, the better the model performance [35,36]. All R codes used for analysis can be obtained from GitHub (https://github.com/RayLing88/Ninghang_SDM (accessed on 11 March 2021)).

The impact of sampling bias on species distribution modeling must be noted [37]. Sampling intensity varies between sites. Oversampling in some geographic spaces can cause repetition in the ecological space when building the model; this affects the simulation of the ecological needs of the species. This kind of sampling bias generally causes overfitting of niche models to species requirements, thus reducing model transferability [38]. To counter sampling bias, we randomly generated 1000 pseudo-absence points throughout China, using Arc-GIS v. 10.2 [39].

The output layers were imported into Arc-GIS v.10.2 for further analysis, and the final distribution map was generated. Using occurrence records from different sources may cause some sampling bias. A tenth-percentile training presence logistic threshold was adopted to define the minimum probability of a suitable habitat for *T. klimeschi* [40]. Based on this, habitat suitability of *T. klimeschi* was divided into four levels: unsuitable (0–0.1), poorly suitable (0.11–0.3), moderately suitable (0.31–0.6), and highly suitable (0.61–1). We then calculated the suitable areas under the future climate scenarios by multiplying the number of presence grid cells by their spatial resolution.

### 2.5. The Centroid Changes Using SDM Tool-Box

We used the SDM tool (http://www.sdmtoolbox.org (accessed on 11 March 2021)) to calculate distributional changes between two binary SDMs (i.e., the current and future SDMs). Centroid analysis summarizes the core distributional shifts in many species’ ranges, and reduces each species’ distribution to a single central point (a centroid); it then creates a vector file depicting the magnitude and direction of the change over time [41].

## 3. Results

### 3.1. Model Performance and Variables’ Response Curves

The model prediction showed a good performance, as the AUC values from the training and test datasets were 0.898 and 0.876, respectively. Based on the regression coefficients and percentage contributions of the environmental variables used in the model, the top five variables—mean temperature of coldest quarter (Bio11; −0.8217, 36.3%), precipitation of wettest month (Bio13; 0.5534, 25.1%), mean temperature of warmest quarter (Bio10; −0.4035, 17.7%), mean temperature of driest quarter (Bio9; −0.1864, 9.6%), and mean diurnal range (Bio2; −0.0569, 7.7%)—were selected as the important factors determining the habitat suitability of *T. klimeschi* (Table 3).

Response curves show the quantitative relationship between environmental variables and the habitat suitability. According to the response curves (Figure 3), we obtained the thresholds (existence probability > 0.3) for the five variables. Mean diurnal range ranged (Bio2) from 11.7 °C to 16.4 °C, mean temperature of driest quarter (Bio9) ranged from −12.3 °C to 2.6 °C, mean temperature of warmest quarter (Bio10) ranged from 12.6 °C to 26.7 °C, mean temperature of coldest quarter (Bio11) ranged from −14.8 °C to −4.2 °C, and precipitation of wettest month (Bio13) ranged from 5 to 42 mm.

### 3.2. Current Potential Distribution of T. Klimeschi

The current potential distribution of *T. klimeschi* is in South Xinjiang and northern Gansu, amounting to 25.58 × 10^4^ km^^2^^ (Figure 4). The total areas for highly, moderately and poorly suitable areas are 2.56 × 10^4^ km^2^, 6.91 × 10^4^ km^2^ and 15.35 × 10^4^ km^2^, respectively. As shown in Figure 2, the suitable area in South Xinjiang is distributed between the southern edge of the Tianshan Mountains and the northern edge of the Tarim Basin, amounting to 20.36 × 10^4^ km^^2^^. In northern Gansu, this beetle is distributed in Dunhuang. Two other potential areas for this species are in or near the cities of Alaer and Korla. The highly suitable habitats occur in discontinuous patches, and the moderately suitable habitats are embedded in these patches.

### 3.3. Future Potential Distribution of T. Klimeschi

Under future climate scenarios, the potential distribution of *T. klimeschi* is projected to increase markedly over time (Figure 5a–f, Figure 6). The response was strongest under the RCP 8.5–2050s climate scenario, with the area increasing to 36.7 × 10^4^ km^2^, followed by the RCP 4.5–2050s climate scenario. Compared with the current distribution, the highly suitable area in Alaer still exists in this future scenario. However, the highly suitable area in Korla will change to a moderately suitable area. Under all assumptions, Jiuquan in Gansu is projected to become a suitable area for *T. klimeschi* in the 2030s. Then, *T. klimeschi* is expected to enter western Inner Mongolia along the Hexi Corridor in the 2050s. In southeastern Xinjiang, however, the suitable area in northern Ruoqiang and most of Korla is projected to shrink.

Under the RCP 2.6–2030s scenario, the suitable area in Dunhuang is projected to expand. By the 2050s, the suitable area is projected to increase continually in Jiuquan. However, the moderately and highly suitable areas in northern Ruoqiang, and most of the area of Korla are projected to decrease persistently. Under the RCP 4.5–2030s scenario, *T. klimeschi* will continue to expand along the Hexi Corridor. By the 2050s, the population in Jiuquan is projected to spread to the junction of Inner Mongolia and Gansu Province. Under the RCP 8.5–2030s scenario, *T. klimeschi* is expected to expand to most regions of western Inner Mongolia. By the 2050s, *T. klimeschi* will gain the largest climatically suitable space. Overall, in the near future, *T. klimeschi* will gain an increasingly suitable climatic niche in northwest China.

### 3.4. The Distributional Centroid Changes

Distributional changes in *T. klimeschi* climate niches are shown in Figure 7. The current distributional centroid is located in Alaer (81°33′ E, 40°57′ N), South Xinjiang. Under the RCP 2.6 scenario, the population centroid will move northeast, to Bugur (84°42′ E, 41°44′ N) by the 2030s, then southwest to Awat (80°27′ E and 40°47′ N) by the 2050s. Under the RCP 4.5 scenario, the centroid will move northeast to Hami, (93°37′ E, 40°59′ N) by the 2030s and then southeast to Dunhuang (94°14′ E, 40°18′ N) by the 2050s. Under RCP 8.5 scenario, the centroid will move southeast to Dunhuang (94°54′ E, 40°25′ N) by the 2030s, and southeast to Jinta (98°49′ E, 40°14 ′ N) by the 2050s.

## 4. Discussion

In northwest China, many *P. alba* var. *pyramidalis* stands are being destroyed by *T. klimeschi.* Previous research on this beetle has focused on integrated control. However, due to its effective concealment, measures to prevent and control outbreaks have not been fully effective. Our current SDM analysis makes it possible to classify and monitor its habitat in a targeted way, based on habitat suitability maps. This both reduces the need for service staff, materials, and funding, and improves the efficiency of monitoring work.

The results showed that *T. klimeschi* occurs between the southern edge of the Tianshan Mountains and the northern edge of the Tarim Basin. In other words, in south Xinjiang, the southern edge of the Tianshan Mountains and the northern edge of the Tarim Basin are the northernmost and southernmost boundaries of the geographical distribution of *T. klimeschi*, respectively. In addition, Dunhuang in northern Gansu also provides a suitable climate that supports the survival of this beetle. Furthermore, it is restricted to this area, despite there being large numbers of hosts in northern China, because the climate of this area is suitable for it. Under all of the climate scenarios that we studied, the suitable area in northern Gansu is projected to expand continuously. Along the Hexi Corridor, *T. klimeschi* could spread into western Inner Mongolia. Jiuquan will develop into the next center of the spread. Although the scale of the changes in suitable areas varied among the scenarios, all of the scenarios suggest difficulties for forest managers in the coming decades. To address this, better monitoring and management are required in these areas. For instance, regular surveys should be conducted to ensure early detection of outbreaks, and vulnerable forest areas should be identified from maps of projected suitable distributions. Pest populations should be carefully monitored, using both visual inspection and trapping systems; this will help to determine when pest-control activity is needed. Robust forest monitoring and reporting systems should be established, to ensure timely warnings of the effects of climate change on pests and hosts, and to measure the effectiveness of management responses. These procedures are an important step in developing management strategies that integrate monitoring systems and projected climate change impacts when conducting vulnerability and risk assessments.

In this study, temperature and precipitation were selected as the variables that constrain the current distribution of *T. klimeschi*; the mean temperature of the coldest quarter is particularly important. The complex relationships between temperature and physiological processes affect species’ geographic distributions [42]. In particular, overwinter survival is a dominant factor limiting the distribution of insects. *T. klimeschi* relies on larvae for overwintering survival [6]. The ability to supercool is a key survival indicator for species living at low temperatures. Among its life-cycle stages, the larvae are considered the most tolerant to cold. It has been reported that −13°C caused more than 90% larvae mortality [5]. Observations of field populations suggest that temperatures below −15 °C reduced larvae survival [5,6]. For many species of bark beetle, synchronous adult emergence and life-cycle timing are required to kill the host trees [43]. Synchronous adult emergence is an important regulator of insect seasonality and synchrony, and ultimately of the mean fitness of the population. To improve their chances of surviving adverse conditions such as extreme cold or heat, vulnerable life-cycle stages must be synchronized with the appropriate seasons (the phenomenon known as seasonality). The complex relationships between temperature and the physiological processes involved in phenology affect bark beetle population dynamics and distribution in many ways. Mechanisms promoting seasonality are critical to bark beetle population growth and outbreak potential [44]. *Trypophloeus klimeschi* adult infestations begin in mid-May and reach a peak from late May to mid-June, completing two generations per year [6]. During adult activity, the extreme variation of diurnal temperature and scarce rainfall in summer (for instance, in the interior of Tarim Basin) are likely to affect its seasonality. Disrupted synchronization can severely impact the growth and development of the insect, even leading to death [45]. The climatic conditions between the southern edge of the Tianshan Mountains and northern edge of the Tarim Basin, which support the presence of this bark beetle, are unique to these areas. These specific temperature and precipitation conditions constrain this species to this region.

*Trypophloeus klimeschi* is absent in most regions of Xinjiang, where *P. alba* var. *pyramidalis* grows. The host trees in the vast northwest of China are distributed far more widely than the beetle. The northward or southward expansion of the suitable habitat, especially in Xinjiang, did not occur in our projections. The results indicated that the distribution of the beetle is not limited to the host’s distribution. Considering the mean temperature of the coldest quarter as the only limiting factor, the southward expansion of *T. klimeschi* is likely to occur. However, the Tarim Basin in the south Xinjiang has the largest Taklimakan Desert in China, where there is no water, and this limits the survival of the host trees. When it comes to the northern limit of the beetle distribution, the Tianshan Mountains are the natural geographical barrier prevent the beetle from spreading northward. Moreover, the mean temperature of the coldest month in northern Xinjiang (−20°C) might not allow the overwintering of this beetle according to its cold resistance. Hence, intertwined factors restrict the southward or northward spread of the beetle. Our findings are similar to those reported for bark beetle species in North America. For example, the northward movement of *Dendroctonus frontalis* in the United States was constrained by the minimum annual temperature isotherm [46]. In western Canada, the distribution of *Dendroctonus ponderosae*, the mountain pine beetle, was constrained by minimum winter temperature [47]. In the Sierra Madre Occidental, Mexico, the *Dendroctonus rhizophagus* distribution was limited by the maximum temperature isotherm [48]. Furthermore, the host tree distribution will respond dynamically to climate change; forest trees may persist via migration, adapt to the new conditions, or go locally extinct [49]. All three scenarios may occur in the host’s response to climate change. However, due to the fact that *P. alba* var. *pyramidalis* is the most important afforestation tree species in Northwest China, we believe that its distribution will continue to expand. The range of *T. klimeschi* may change to track these environmental changes. Although our results suggested alterations in the potential distribution of *T. klimeschi*, they did not provide absolute predictions. So far, we have not found that this bark beetle feeds on other host trees. We only considered the influence of *P. alba* var. *pyramidalis* distribution on *T. klimeschi* distribution. Therefore, there are still various uncertainties. For example, if the host species of the beetle become more diverse in the future climate, the current projections may prove to be conservative. However, climate change will affect bark beetle–host interactions in complex and nonlinear ways [50,51]. Therefore, future studies should consider the impacts of climate change on the host trees when studying the effects of climate change on the distribution of this bark beetle.

There are some limitations in our study. First, our occurrence data were relatively small. Although the Maxent model has shown advantages in terms of predictive performance for use with a small sample size, the sample size may still affect the accuracy of the model results [52]. Second, only 23 variables were used to model the potential distribution of *T. klimeschi*. The model is based on an ideal niche without considering the effects of biotic factors such as dispersal, geographic barriers, competition, predation, and herbivory, which often also play roles in determining species distributions, which is clearly a critical limitation. Finally, despite the individual SMD model in this study showing a high prediction accuracy, there are still some limitations in the precision of prediction. The accuracy and performance of individual SMDs vary widely among methods and species [53,54]. Some studies have showed that methods integrating multiple individual models provide robust estimates of potential species’ distributions, providing a way to improve the accuracy of models [54,55]. All of the above factors may cause differences between the predicted distributions and actual distributions.

Maxent models describe the association between the occurrence of species populations and environmental variables [23]. We found that bark-beetle-induced tree mortality was correlated with the climate variables describing conditions during the mortality events. Such analyses are usually retrospective, rather than describing the processes leading to plant mortality [56]. Moreover, because these relationships are likely to be different under future climatic conditions, ecosystem models must include phenology data if they are to explain physiological changes in response to a changing climate. The effects of temperature on insect physiology have been studied for decades, with the primary focus being on how temperature affects development time and survival, and how photoperiod and temperature affect diapause [57,58]. Phenological models are driven by the functional relationships between physiological processes and temperature, rather than by statistical relationships [59]. Models describing bark beetle phenology require detailed information on the responses of individual beetles. Globally, there are at least 30 bark beetle species that can cause landscape-scale plant mortality. Of these, sufficient data to model climate-driven phenology are available for only six species, including *D. frontalis*, *D. ponderosa*, and *D. rufipennis* [43]. Therefore, in order to model the phenology of *T. klimeschi*, we first need to collect detailed physiological information for it. Using these data, we will be able to use a phenological model to predict how temperature will impact life-cycle timing and ultimately population success in this species, based on a quantitative description of the physiological processes that are impacted by temperature.

## 5. Conclusions

Temperature and precipitation constrain the current distribution of *T. klimeschi*. With climate change, the eastward expansion of its suitable area will occur. The impacts of climate change will also increase over time. By the 2050s, the total area suitable for this species under the RCP 8.5 scenario will reach 36.7 × 10^4^ km^2^. More and more shelter forests will be invaded by this beetle. Therefore, pest monitoring and control measures should be taken to prevent it from spreading further eastward.

## Figures and Tables

**Figure 1 insects-12-00242-f001:**
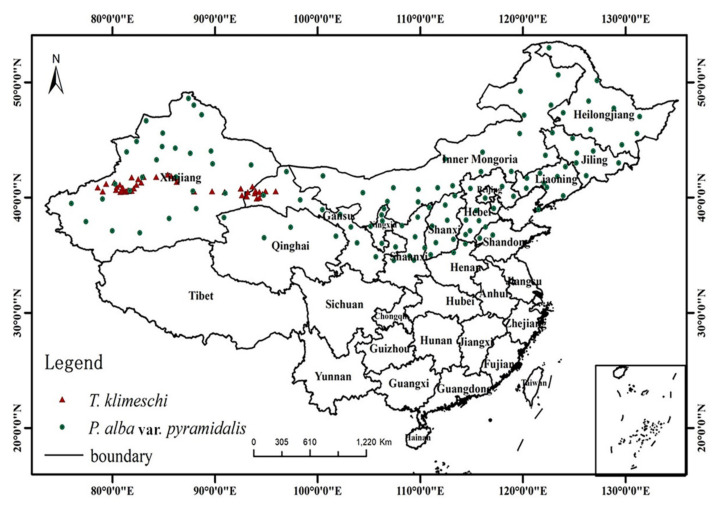
Occurrence records of *T. klimeschi* and current distribution *P. alba* var. *pyramidalis* in China.

**Figure 2 insects-12-00242-f002:**
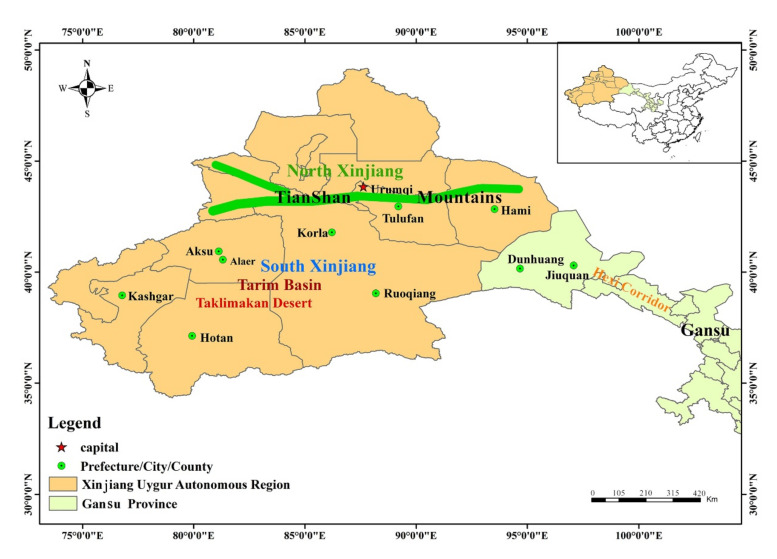
Main geographical locations and features of occurrence area.

**Figure 3 insects-12-00242-f003:**
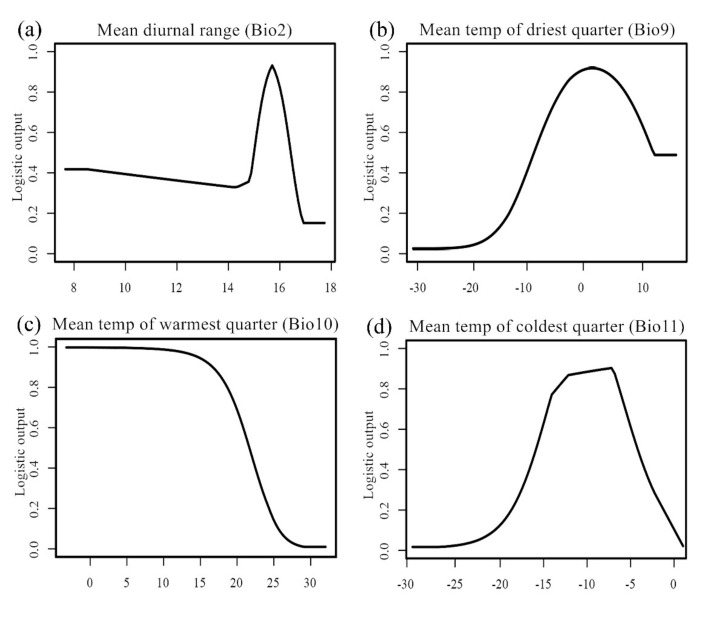
Response curves for dominant environmental variables.

**Figure 4 insects-12-00242-f004:**
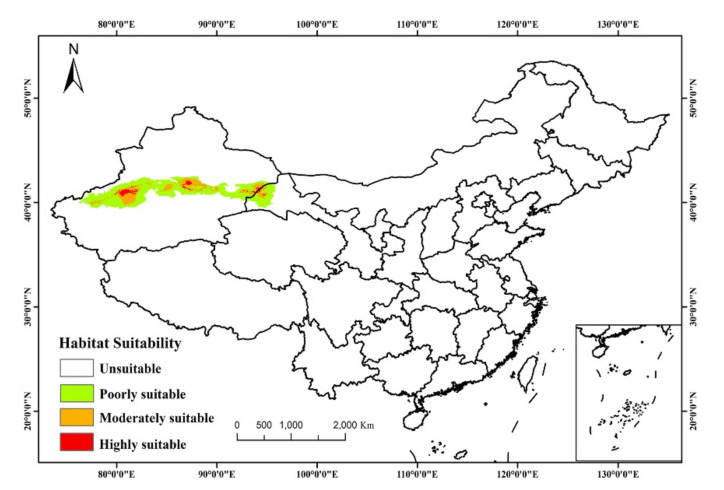
Present habitat distribution suitability of *T. klimeschi*.

**Figure 5 insects-12-00242-f005:**
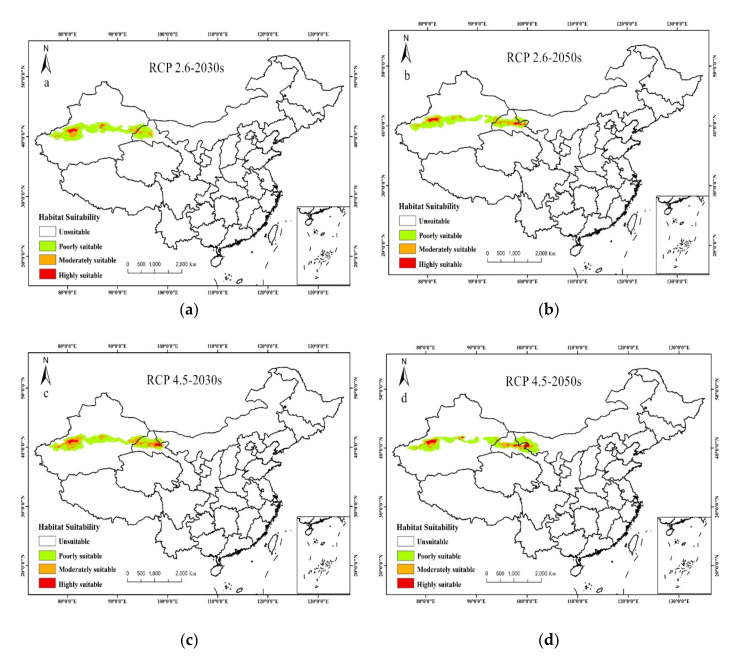
Future habitat distribution suitability of *T. klimeschi*. (**a**,**b**) future suitable habitats under RCP2.6 in 2030s and 2050s; (**c**,**d**) future suitable habitats under RCP4.5 in 2030s and 2050s; (**e**,**f**) future suitable habitats under RCP8.5 in 2030s and 2050s).

**Figure 6 insects-12-00242-f006:**
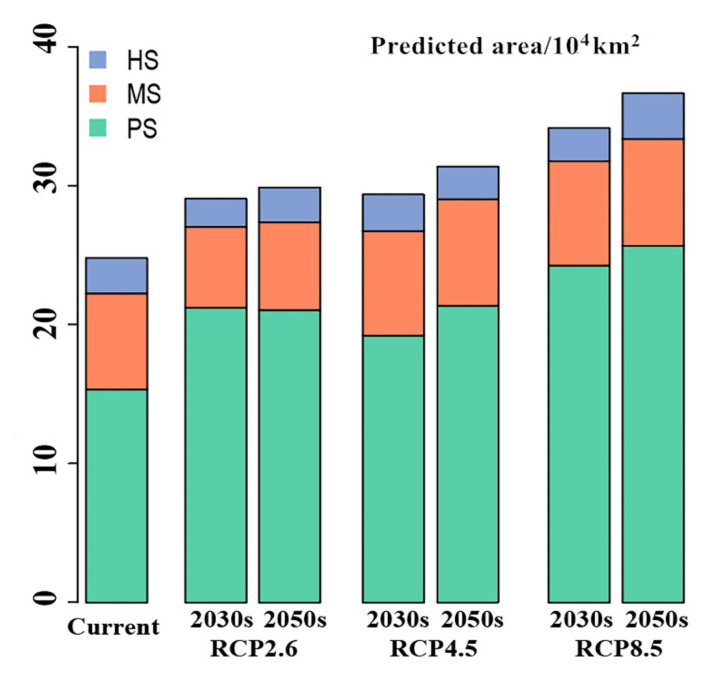
Predicted suitable areas for *T. klimeschi* under current and future climatic conditions (PS represents poorly suitable; MS represents moderately suitable; HS represents highly suitable).

**Figure 7 insects-12-00242-f007:**
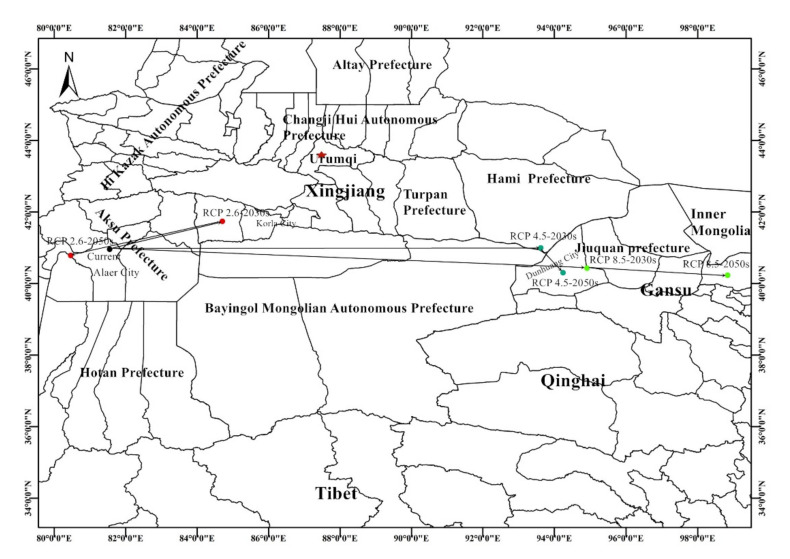
The core distributional shifts of *T. klimeschi* (Black dot represents current centroid; Red dot represents RCP 2.6 centroid; Bottle-green dot represents RCP 4.5 centroid; Bright-green dot represents RCP 4.5 centroid).

**Table 1 insects-12-00242-t001:** Profile of the cultivation area of *Populus alba* var. *pyramidalis.*

Climate Belt	Environmental Overview	Distribution Records
Northern warm temperate deciduous broad-leaved forest	The annual average temperature is 7–12 °C, the average temperature of the coldest month is −10∼−3 °C, the average temperature of the warmest month is 18∼27 °C, and the annual precipitation is 500–700 mm.	Shenyang, Huludao, Dalian, Dandong, Anshan, Liaoyang, Jinzhou, Yingkou, Panjin, Beijing, Tianjin, Taiyuan, Linfen, Changzhi, Shijiazhuang, Qinhuangdao, Baoding, Tangshan, Handan, Xingtai, Chengde, Jinan, Dezhou, Yan’an, Baoji, Tianshui
Temperate grassland	The annual average temperature is −3–−9 °C, the accumulated temperature of ≥ 10 °C is 1600–3200 °C, and the average temperature of coldest month is −7–−29 °C. The annual precipitation is 150–500 mm, mostly below 350 mm, mainly in summer.	Lanzhou, Pingliang, Altay, Hailar, Manzhouli, Qiqihar, Fuxin, Dandong, Daqing, Xining, Yinchuan, Tongliao, Yulin, Hohhot, Baotou, Zhangjiakou, Jining, Chifeng, Datong, Xilinhot
Temperate desert	This climate belt is distributed mainly in Xinjiang. The annual average temperature is 4–9 °C in northern Xinjiang and 7–14 °C in southern Xinjiang. The average temperature in January is −20–15 °C in northern Xinjiang and −10 °C–5 °C in southern Xinjiang. Most of the annual rainfall is below 50–100 mm, and the least is only 10–20 mm.	Urumqi, Shihezi, Karamay, Hami Kashgar, Wuwei, Jiuquan, Yumen, Jiayuguan, Golmud, Korla, Jinchang, Wuhai

**Table 2 insects-12-00242-t002:** Description of environmental variables used for modeling.

Data Source	Category	Environmental Variables (unit)	Abbreviation
WorldClim	Bioclimatic	Annual mean temperature (°C)	Bio1
		Mean diurnal range (°C)	Bio2
		Isothermality (%)	Bio3
		Temperature seasonality (°C)	Bio4
		Maximum temperature of warmest month (°C)	Bio5
		Minimum temperature of coldest month (°C)	Bio6
		Temperature annual range (°C)	Bio7
		Mean temperature of wettest quarter (°C)	Bio8
		Mean temperature of driest quarter (°C)	Bio9
		Mean temperature of warmest quarter (°C)	Bio10
		Mean temperature of coldest quarter (°C)	Bio11
		Annual precipitation (mm)	Bio12
		Precipitation of wettest month (mm)	Bio13
		Precipitation of driest month (mm)	Bio14
		Precipitation seasonality	Bio15
		Precipitation of wettest quarter (mm)	Bio16
		Precipitation of driest quarter (mm)	Bio17
		Precipitation of warmest quarter (mm)	Bio18
		Precipitation of coldest quarter (mm)	Bio19
USGS	Terrain	Altitude (m)	Alt.
		Aspect (degree)	Asp.
		Slope (degree)	Slop.
GBIF, CVH, Field investigations	Host	*P. alba* var. *pyramidalis* distribution	H

**Table 3 insects-12-00242-t003:** Ranking of the importance of variables for prediction of the distribution of *T. klimeschi.*

Rank	Environmental Variables	Regression Coefficients in LASSO	Contribution (%)	Probability of Selection
1	Mean temperature of coldest quarter	−0.8217	36.3	1.00
2	Precipitation of wettest month	0.5534	25.1	0.98
3	Mean temperature of warmest quarter	−0.4035	17.7	0.96
4	Mean temperature of driest quarter	−0.1864	9.6	0.93
5	Mean diurnal range	−0.0569	7.4	0.92

## Data Availability

Author suggest to exclude this statement.

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
