# Peer review of "Mapping Invasion Potential of the Pest from Central Asia, Trypophloeus klimeschi (Coleoptera: Curculionidae: Scolytinae), in the Shelter Forests of Northwest China"

_insects, 2021, doi:10.3390/insects12030242_

Round 1

Reviewer 1 Report

The work of Ning et al. entitled “Mapping invasion potential of the pest from Central Asia, Trypophloeus klimeschi (Coleoptera: Curculionidae: Scolytinae), in the shelter forests of Northwest China” focused on the application of Species Distribution Models (SDMs) to map the actual distribution of T. klimeschi and potential spread under future climate change scenarios. Although the manuscript covers a very interesting topic and authors conducted correctly the analyses, there are several issues to be discussed. In addition, an English language revision is necessary. For these reasons, I suggest to accept the manuscript only after a major revision. Detailed comments and suggestions are reported below:

Line 13: What does it mean “invasive pest”? A pest should be considered as invasive mainly outside from its native range.

Line 14: Does it cause damage only to healthy trees? Authors must provide specific references to confirm this.

Line 40: Authors did not include information about native distribution, as well as its biology and damage-tree association.

Lines 55-56: What does “short” mean? Some species complete up to 10 generations per year, others need more than 4-5 years. Moreover, even if insects are poikilotherm organisms, the effect of temperature on insects vary depending on species. Please correct the sentence avoiding being too much speculative.

Lines 58-59: What was that rate referred to? Reference authors used to declare it resulted difficult to be obtained. Please include more information about it.

Line 60-61: Since new records has been reported, please include specific and recent references.

Line 62: Here and elsewhere within the manuscript, “eruption” must be corrected with “spread”.

Lines 134-136: This information is unnecessary. Please delete the sentence.

Lines 138-140: Correct the sentence by reversing the parts as “To determine the future distributions of T. klimeschi under different climate scenarios, dataset on future climate from … were used.”.

Lines 163-168: Hoh was the randomization performed? Moreover, dividing 89 occurrence points into 10 subsets doesn’t seem completely corrected due to an excessive reduction of the starting dataset. Why did authors do this? Please, include more specification and, if possible, references to justify this approach.

Line 203: AUC value of 0.792 do not represent an “excellent” performance but can be considered as good. Please correct the sentence accordingly. In addition, authors must use past tense rather than present one (i.e. “was” rather than “is”).

Line 205: What does “important” mean? Maybe “most contributing”? This must be corrected or well explained in the text and in Table 3 caption.

Lines 211-217: How did authors extrapolate this information from curves? A more detailed explanation is necessary. Moreover, authors must include acronyms of each variable in the text in order to associate the text with the reference figure.

Line 224: Include description of acronyms BIO-XX reported in the figure caption to make easier the comprehension of the figure.

Lines 226-234: The entire paragraph should be rewritten as is particularly difficult to be understand. In fact, readers are not required to know China geographic characteristics. However, authors could leave the text in its current form including references reported in the text into the figure.

Line 257: Please substitute the Figure 5 with a higher quality image.

Lines 260-269: As already reported in a previous comment, authors must either report more details in the figures or rewrite completely the paragraph. Potential readers may not know China localities and municipalities.

Lines 299-300: Since authors wrote that “the climate of this area is suitable for it”, a better explanation on the most climatic suitable condition for T. klimeschi is needed.

Line 302: Substitute “is expected to” with “could”.

Lines 319-325: Although information authors reported can be considered corrected, they did not argue sufficiently how temperatures limit the range of expansion of the pest in the specific case. A better explanation is needed.

Line 333: Here and elsewhere in the manuscript, substitute “T. klimeschi” with “Trypophloeus klimeschi” when used in the beginning of a sentence.

Line 356: substitute “T. klimesch” with “T. klimeschi”.

Author Response

Dear Reviewer:

Thank you for your letter and for the reviewers’ comments concerning our manuscript entitled “Mapping invasion potential of the pest from Central Asia, Trypophloeus klimeschi (Coleoptera: Curculionidae: Scolytinae), in the shelter forests of Northwest China” (ID: insects-118240). Those comments are all valuable and very helpful for revising and improving our paper, as well as the important guiding significance to our researches. We have studied comments carefully and have made correction which we hope meet with approval. Revised portion are marked in red and added portion are marked in blue in the manuscript.

The main corrections in the manuscript and the responds to the reviewer’s comments are as flowing:

The work of Ning et al. entitled “Mapping invasion potential of the pest from Central Asia, Trypophloeus klimeschi (Coleoptera: Curculionidae: Scolytinae), in the shelter forests of Northwest China” focused on the application of Species Distribution Models (SDMs) to map the actual distribution of T. klimeschi and potential spread under future climate change scenarios. Although the manuscript covers a very interesting topic and authors conducted correctly the analyses, there are several issues to be discussed. In addition, an English language revision is necessary. For these reasons, I suggest to accept the manuscript only after a major revision. Detailed comments and suggestions are reported below:

1. Line 13: What does it mean “invasive pest”? A pest should be considered as invasive mainly outside from its native range.

Re: We have revised it and added more contents about the pest in background according to your suggestion. “Trypophloeus klimeschi Eggers (Coleoptera: Curculionidae: Scolytinae) is a newly recorded species in the Xinjiang Uygur Autonomous Region, China. T. klimeschi is native to the Kyrgyz Republic, which borders Xinjiang.”

2. Line 14: Does it cause damage only to healthy trees? Authors must provide specific references to confirm this.

Re: We have revised it in simple summary and background according to your suggestion. Meanwhile, the same problem throughout the manuscript has been corrected.

Trypophloeus klimeschi Eggers (Coleoptera: Curculionidae: Scolytinae) causes substantial mortality to Populus alba var. pyramidalis individuals in Xinjiang, China.”

3. Line 40: Authors did not include information about native distribution, as well as its biology and damage-tree association.

Re: We have added the contents in background according to your suggestion.

Introduction:

Trypophloeus klimeschi Eggers (Coleoptera: Curculionidae: Scolytinae) is a newly recorded species in China. T. klimeschi is native to the Kyrgyz Republic, which borders Xinjiang Uygur Autonomous Region. In 2003, the beetle was first found in Aksu prefecture, Xinjiang. Subsequently, the beetle spread rapidly to adjacent areas and is now found in Dunhuang city, Gansu Province. T. klimeschi first invades the branch shoots of Populus alba var. pyramidalis Bunge and then gradually spreads to the main trunk. The beetle completes its life cycle in the phloem under the bark, except during a brief dispersal when the adults search for new host trees. The injured branches turn yellow and wither, many of the leaves fall off, and dense holes are formed in the trunk surface, causing the injured trees to wither and rapidly die. T. klimeschi typically has two generations per year and mature larvae overwinter. First-generation adults emerge from late-May to late-June and second-generation adults emerge in August. T. klimeschi had strong host-selection specificity, which can cause the extent of injury varies with age of P. alba var. pyramidalis. However, other local poplar species, such as P. alba L., P. tomentosa Carr, and P. dakuanensis Hsu, are not invaded by this beetle. In recent years, a large number of host trees in northwestern China were killed by this beetle, which overburdened the original fragile shelterbelt. As a preferred afforestation tree species for greening and shelterbelts, especially in the northern China, many P. alba var. pyramidalis stands provide an ecological corridor for the spread of T. klimeschi from the northwestern to the eastern region of China.

4. Lines 55-56: What does “short” mean? Some species complete up to 10 generations per year, others need more than 4-5 years. Moreover, even if insects are poikilotherm organisms, the effect of temperature on insects vary depending on species. Please correct the sentence avoiding being too much speculative.

Re: We have made revised according to your comments. “Insects are poikilotherms, influenced by changes of temperature, and the increase in global temperature will change the growth and development rate of insects, the number of populations produced, and their geographical distribution.”

5. Lines 58-59: What was that rate referred to? Reference authors used to declare it resulted difficult to be obtained. Please include more information about it.

Re: We have made revised it according to your comments. Because the reference of the rate is not clear, and the sample site of the survey only occurs in Dunhuang. It's confusing, so we delete the sentence.

6. Line 60-61: Since new records has been reported, please include specific and recent references.

Re: We have made added the references in the manuscript according to your comments.

7. Line 62: Here and elsewhere within the manuscript, “eruption” must be corrected with “spread”.

Re: We have made revised according to your comments. In addition, we also check and revise the same question throughout the manuscript.

8. Lines 134-136: This information is unnecessary. Please delete the sentence.

Re: We have made deleted it in the manuscript according to your comments.

9. Lines 138-140: Correct the sentence by reversing the parts as “To determine the future distributions of T. klimeschi under different climate scenarios, dataset on future climate from … were used.”.

Re: We have made revised it in the manuscript according to your comments. “To determine the future distributions of T. klimeschi under different climate scenarios, dataset on future climate from the Climate Change, Agriculture and Food Security program (CCAFS, http://ccafs-climate.org/) were used.”

10. Lines 163-168: Hoh was the randomization performed? Moreover, dividing 89 occurrence points into 10 subsets doesn’t seem completely corrected due to an excessive reduction of the starting dataset. Why did authors do this? Please, include more specification and, if possible, references to justify this approach.

Re: We have made added reference in the manuscript according to your comments. As you mentioned, dividing 89 occurrence points into 10 subsets doesn’t seem completely corrected due to an excessive reduction of the starting dataset. Therefore, we choose a better division When running testing dataset and training dataset independently. Usually, the data set is divided according to 70% or 75% (training dataset). However, we found that this division is not good in this fitting procedure. Therefore, we firstly divide the dataset into 10 subsets. Then, the fitting effect with the division of 90% is better than other divisions. The evaluation and selection of the model correspond to the determination of the predictive capacity based on the quantification of the error and the misclassified data. We randomly split our dataset into two subsets, 90% of the data to train the model and 10% for model evaluation using cross-validation (100) which yielded 100 different fits per model. Models with higher mean values and smaller variations were considered as being the most accurate ones. In addition, Lasso regression is one of the regularization methods that creates parsimonious models in the presence of collinearly among variables. Lasso regression performs L1 regularization that is it adds the penalty equivalent to the absolute value of the magnitude of the coefficients. Here the minimization objective is as followed.

Minimization objective = LS Obj + λ (sum of absolute value of coefficients)

Cross validation strategy based on glmnet is applied to find the optimum λ. Once lambda is determined, then we can get the best model.

11. Line 203: AUC value of 0.792 do not represent an “excellent” performance but can be considered as good. Please correct the sentence accordingly. In addition, authors must use past tense rather than present one (i.e. “was” rather than “is”).

Re: We obtained five-number summary of AUC value based on 100-times cross-validation. Among the five-number summary, 0.792, 0.898, 0.968 is the minimum, median, and maximum of the data set. Therefore, we have made revised in the manuscript according to your comments.

“The AUC values of training dataset and test dataset respectively was 0.898 and 0.876, showing the model performance was good.”

12. Line 205: What does “important” mean? Maybe “most contributing”? This must be corrected or well explained in the text and in Table 3 caption.

Re: We have made revised in the manuscript and Table 3 caption according to your comments.

In selection of variables, we used the maximum entropy gain and least absolute shrinkage method to selected important variables. The variables with higher maximum entropy gain and non-zero regression coefficients were used to the important variables. Therefore, the importance variables, selected though multicollinearity test, were used in modelling. “Based on the regression coefficients and percentage contribution of the environmental variables used in the model, the top five variables, including mean temperature of coldest quarter (Bio11; -0.8217, 36.3%), precipitation of wettest month (Bio13; 0.5534, 25.1%), mean temperature of warmest quarter (Bio10; -0.4035, 17.7%), mean temperature of driest quarter (Bio9; -0.1864, 9.6%), mean diurnal range (Bio2; -0.0569, 7.7%) were selected as the important factors determining the habitat suitability of T. klimeschi (Table 3).”

Table 3. Ranking of the importance of variables for prediction of the distribution of T. klimeschi

Rank

Environmental variables

Regression coefficients

in LASSO

Contribution (%)

Probability of selection

1

Mean temperature of coldest quarter

-0.8217

36.3

1.00

2

Precipitation of wettest month

0.5534

25.1

0.98

3

Mean temperature of warmest quarter

-0.4035

17.7

0.96

4

Mean temperature of driest quarter

-0.1864

9.6

0.93

5

Mean diurnal range

-0.0569

7.4

0.92

The importance variables, selected though multicollinearity test, were used in modelling.

13. Lines 211-217: How did authors extrapolate this information from curves? A more detailed explanation is necessary. Moreover, authors must include acronyms of each variable in the text in order to associate the text with the reference figure.

Re: We have added selection criteria in the manuscript. “According to the response curves (Figure 2), we obtained the thresholds (existence probability > 0.3) for the five variables.” Besides, acronyms of each variable were also added.

14. Line 224: Include description of acronyms BIO-XX reported in the figure caption to make easier the comprehension of the figure.

Re: We have revised in the figure.

15. Lines 226-234: The entire paragraph should be rewritten as is particularly difficult to be understand. In fact, readers are not required to know China geographic characteristics. However, authors could leave the text in its current form including references reported in the text into the figure.

Re: We have added in the figure 2 and identified it.

Figure 2. Main geographical locations and characteristics of occurrence area.

16. Line 257: Please substitute the Figure 5 with a higher quality image.

Re: We have substituted the Figure 5 with a higher quality image.

17. Lines 260-269: As already reported in a previous comment, authors must either report more details in the figures or rewrite completely the paragraph. Potential readers may not know China localities and municipalities.

Re: We have added in the figure and identified it.

18. Lines 299-300: Since authors wrote that “the climate of this area is suitable for it”, a better explanation on the most climatic suitable condition for T. klimeschi is needed.

Re: We have made explanations in the manuscript caption according to your comments. In the discussion of host distribution, we added the following: Trypophloeus klimesch was absent in the most region of Xinjiang where P. alba var. pyramidalis exists. The host trees in the vast northwest of China are distributed far more widely than the beetle. The north or southward expansion of the suitable habitat, especially in Xinjiang, was not occurred in our projections. The resulted indicated the distribution of the beetle is not limited by the host’s distribution. Only considering that the mean temperature of coldest quarter is limiting factor, southward expansion of T. klimeschi is likely to occur. However, the Tarim Basin in the south Xinjiang has the largest Taklimakan Desert in China, where there is no water and limits the survival of the host trees. As the herbivorous insect, T. klimeschi can’t continue to occur in those places where their host plants are not present. When it comes to the northern limit of the beetle distribution, Tianshan Mountains is the natural geographical barrier to prevent the beetle from spreading northward. Moreover, the mean temperature of the coldest month in northern Xinjiang generally can reach - 20°C can’t guarantee the overwintering of this beetle according its cold resistance. Hence, the intertwined factors restrict the south or northward spread of the beetle.

19. Line 302: Substitute “is expected to” with “could”.

Re: We have made revised it in the manuscript according to your comments.

20. Lines 319-325: Although information authors reported can be considered corrected, they did not argue sufficiently how temperatures limit the range of expansion of the pest in the specific case. A better explanation is needed

Re: We have added the contents in the manuscript according to your comments. In this study, the mean temperature of coldest quarter is the most important limiting factor. As the previous question, "The climate of this area is suitable for it", we highlight the uniqueness of the occurrence area of the beetle. If the mean temperature of coldest quarter is limiting factor, southward expansion of T. klimeschi is likely to occur. Combined with host distribution and characteristics, a better explanation follows. The Tarim Basin in the south Xinjiang has the largest Taklimakan Desert in China, where there is no water and limits the survival of the host trees. As the herbivorous insect, T. klimeschi can’t continue to occur in those places where their host plants are not present. When it comes to the northern limit of the beetle distribution, Tianshan Mountains is the natural geographical barrier to prevent the beetle from spreading northward. Moreover, the mean temperature of the coldest month in northern Xinjiang generally can reach - 20°C can’t guarantee the overwintering of this beetle according to its cold resistance.

21. Line 333: Here and elsewhere in the manuscript, substitute “T. klimeschi” with “Trypophloeus klimeschi” when used in the beginning of a sentence.

Re: We have made revised in the manuscript according to your comments.

22. Line 356: substitute “T. klimesch” with “T. klimeschi

Re: We have made revised in the manuscript according to your comments.

Thanks for your valuable comments.

Reviewer 2 Report

Manuscript Number: Insects-1118240

General comments

The manuscript entitled “Mapping invasion potential of the pest from Central Asia, Trypophoeus klimeschi (Coleoptera: Curculionidae: Scolytinae), in the shelter forests of Northwest China” dealt with invasive potential of new exotic species in China using SDM(species distribution model). This is interesting topic and has merits of publication. However, I recommended several improvements of the manuscript in aspects of readability and modeling logic.  

1) Justification on expansion of suitable habitat to only eastward is necessary. The host tree was found in the most region of Xinjang and the north or southward expansion of the suitable habitat was not observed in your model. Therefore, host distribution is not limiting factor. If the mean temperature of coldest quarter is limiting factor, southward expansion is expected. I recommended proper discussion of the expansion trends in your model. Probably your sampling or data collection were concentrated on middle of Xinjang and these affects your results. If you can show absence data in north or south Xinjang, your modeling results can be justified. Anyway, the reason for only eastward expansion should be explained in the manuscript in detail.

2) More explanation on field survey of insect collection is necessary. Due to cryptic nature of bark beetle, trapping device generally used for field survey. In this manuscript, field survey process was not described. It is necessary to include the information on sampling in the field such as sampling device, periods for collection and the number of sampling per site or plot. You mentioned that 89 point-based occurrence records for the insect (line 102-116) but only about 20-30 points for the insect was found in Fig. 1. How many points come from field work or literature survey among 89 points? Do you have absence data of the insect through field works? If you have, mark them in Fig. 1.  

3) The information on Trypophoeus klimeschi is necessary in Introduction. From title, I guessed that the pest probably originated from Central Asia but I cannot find any comments in your manuscript. I strongly recommended to include more information on the pest: its geographical distribution, biology, hosts(including whether it is monophagous or polyphagous), and invasive history in China. Geographical distribution of potential host can affect your modeling results. Therefore, basic information for the insect is necessary.

4) In case of invasive species, its dispersal and occurrence were usually affected by human activities. Therefore, distance from road or traffic were included for modelling approaches. Including of human activities in your model probably increase explanatory power of your model.

5) Line 71-72: spell out the name of model.

6) Figures in your manuscript is not easy to read. Please substitute all of figures to high resolution ones. Legends of some figures or table (Fig. 2 etc.) are required more detailed explanation.  

Author Response

Dear Reviewer:

Thank you for your letter and for the reviewers’ comments concerning our manuscript entitled “Mapping invasion potential of the pest from Central Asia, Trypophloeus klimeschi (Coleoptera: Curculionidae: Scolytinae), in the shelter forests of Northwest China” (ID: insects-118240). Those comments are all valuable and very helpful for revising and improving our paper, as well as the important guiding significance to our researches. We have studied comments carefully and have made correction which we hope meet with approval. Revised portion are marked in red and added portion are marked in blue in the manuscript.

The main corrections in the manuscript and the responds to the reviewer’s comments are as flowing:

General comments

The manuscript entitled “Mapping invasion potential of the pest from Central Asia, Trypophoeus klimeschi (Coleoptera: Curculionidae: Scolytinae), in the shelter forests of Northwest China” dealt with invasive potential of new exotic species in China using SDM (species distribution model). This is interesting topic and has merits of publication. However, I recommended several improvements of the manuscript in aspects of readability and modeling logic.

1) Justification on expansion of suitable habitat to only eastward is necessary. The host tree was found in the most region of Xinjiang and the north or southward expansion of the suitable habitat was not observed in your model. Therefore, host distribution is not limiting factor. If the mean temperature of coldest quarter is limiting factor, southward expansion is expected. I recommended proper discussion of the expansion trends in your model. Probably your sampling or data collection were concentrated on middle of Xinjiang and these affects your results. If you can show absence data in north or south Xinjiang, your modeling results can be justified. Anyway, the reason for only eastward expansion should be explained in the manuscript in detail.

Re: We have added the discussion content of the expansion trends according to your suggestion.

Discussion:

Trypophloeus klimeschi was absent in the most region of Xinjiang where P. alba var. pyramidalis exists. The host trees in the vast northwest of China are distributed far more widely than the beetle. The north or southward expansion of the suitable habitat, especially in Xinjiang, was not occurred in our projections. Therefore, T. klimeschi distribution is not limited by the host’s distribution. Only considering that the mean temperature of coldest quarter is limiting factor, population southward expansion is expected. However, the Tarim Basin in the south Xinjiang has the largest Taklimakan Desert in China, where there is no water and limits the survival of the host trees. As the herbivorous insect, T. klimeschi can’t continue to occur in those places where their host plants are not present. When it comes to the northern limit of the beetle distribution, Tianshan Mountains where is a dividing line between north and south Xinjiang are the natural geographical barrier to prevent the beetle from spreading northward. Moreover, according some research results, it has been reported that −13°C caused more than 90% larvae mortality. Observations of field populations suggest that temperatures below -15 °C reduced larvae survival. Therefore, the mean temperature of the coldest month in northern Xinjiang generally can reach - 20°C, which may not guarantee the overwintering of this beetle. Hence, the intertwined factors restrict the south or northward spread of the beetle.

Yes, sampling or data collection were concentrated on middle of Xinjiang and these are likely to affects the results of modelling. We also made some explanations in the manuscript. “The impact of sampling bias on species distribution modeling must be noted. Sampling intensity varies between sites. Oversampling in some geographic spaces can cause repetition in the ecological space when building the model; this will affect simulation of the ecological needs of the species. This kind of sampling bias will generally cause overfitting of niche models to species requirements, thus reducing model transferability. To counter sampling bias, we randomly generated 1,000 pseudo-absence points throughout China, using Arc-GIS v. 10.2.”

In the discussion section of the manuscript, we also emphasize the limitations.

2) More explanation on field survey of insect collection is necessary. Due to cryptic nature of bark beetle, trapping device generally used for field survey. In this manuscript, field survey process was not described. It is necessary to include the information on sampling in the field such as sampling device, periods for collection and the number of sampling per site or plot. You mentioned that 89 point-based occurrence records for the insect (line 102-116) but only about 20-30 points for the insect was found in Fig. 1. How many points come from field work or literature survey among 89 points? Do you have absence data of the insect through field works? If you have, mark them in Fig. 1.

Re: We have added the content of field survey according to your suggestion. According to the infected area, the county forestry departments is regarded as the survey units. The survey route was carried out from west to east along the northern edge of Tarim Basin. Specifically, the Hotan Prefecture in Xinjiang and the Dunhuang City in Gansu were starting and ending of the survey route, respectively. The distance between the investigative points > 20 km. When two or more beetles were observed in close proximity, we recorded only one occurrence. Due to cryptic nature of bark beetle, trapping device generally were used monitor beetle presence in areas with population outbreaks. At each occurrence, collection of the pest is generally counted once a week and the its present or absence was recorded. As for the only 20-30 points in Figure 1, most of the points are concentrated in Aksu area, southern Xinjiang according to the sampling. These points represent the existence of the beetle in the geographical location, but it does not represent that each point represents a record. Among them, some points are close in geographical coordinates. To avoid the occurrences concentration or high coincidence degree, we reduce the occurrence records in the figure. The existing points in the figure are saved at the points with large difference in longitude and latitude. In the manuscripts, we also interpret it as the distance between the investigative points > 20 km. When two or more beetles were observed in close proximity, we recorded only one occurrence. Finally, we obtained 89 point-based occurrence records. There are only seven points obtained through literature, and others are from field survey.

3) The information on Trypophoeus klimeschi is necessary in Introduction. From title, I guessed that the pest probably originated from Central Asia but I cannot find any comments in your manuscript. I strongly recommended to include more information on the pest: its geographical distribution, biology, hosts (including whether it is monophagous or polyphagous), and invasive history in China. Geographical distribution of potential host can affect your modeling results. Therefore, basic information for the insect is necessary.

Re: We have added these contents in background according to your suggestions.

Introduction:

Trypophloeus klimeschi Eggers (Coleoptera: Curculionidae: Scolytinae) is a newly recorded species in China. T. klimeschi is native to the Kyrgyz Republic, which borders Xinjiang Uygur Autonomous Region. In 2003, this beetle was first found in Aksu prefecture, Xinjiang. Subsequently, the beetle spread rapidly to adjacent areas and is now found in Dunhuang city, Gansu Province. T. klimeschi first invades the branch shoots of Populus alba var. pyramidalis Bunge and then gradually spreads to the main trunk. The beetle completes its life cycle in the phloem under the bark, except during a brief dispersal when the adults search for new host trees. The injured branches turn yellow and wither, many of the leaves fall off, and dense holes are formed in the trunk surface, causing the injured trees to wither and rapidly die. T. klimeschi typically has two generations per year and mature larvae overwinter. First-generation adults emerge from late-May to late-June and second-generation adults emerge in August. T. klimeschi had strong host-selection specificity, which can cause the extent of injury varies with age of P. alba var. pyramidalis. However, other local poplar species, such as P. alba L., P. tomentosa Carr, and P. dakuanensis Hsu, are not invaded by this beetle. In recent years, a large number of host trees in northwestern China were killed by this beetle, which overburdened the original fragile shelterbelt. As a preferred afforestation tree species for greening and shelterbelts, especially in the northern China, many P. alba var. pyramidalis stands provide an ecological corridor for the spread of T. klimeschi from the northwestern to the eastern region of China.

4) In case of invasive species, its dispersal and occurrence were usually affected by human activities. Therefore, distance from road or traffic were included for modelling approaches. Including of human activities in your model probably increase explanatory power of your model.

Re: Yes. At the macro level, P. alba var. pyramidalis (the host) are the preferred afforestation tree species for greening and shelterbelts in the North China, especially in the “Three North Area.” Many stands of P. alba var. pyramidalis in northern China provide a continuous corridor for the spread of T. klimeschi from the northwestern to the eastern region of China. Therefore, its dispersal and occurrence were usually affected by human activities. However, the data about distance from road or traffic in the occurrence area is limited due to various reasons, and the distribution of sampling points is uneven. Even if generated background points can reduce this sampling bias, it may still lead to bias. Therefore, we didn’t add the relevant factors of human activities at the beginning of the experiment design. However, I have made the corresponding explanation in the discussion part of the article. Considering that the problem of modelling scale, we though that the analysis used local-scale may be more convincing when it is likely to know the impact of these factors on the beetle. It is not likely to select China or a larger scale (global) to extract more critical information. In the future, we need to collect more data related to human activities to make the prediction more accurate.

5) Line 71-72: spell out the name of model.

Re: We have checked and identified these species distribution models according to your suggestions.

6) Figures in your manuscript is not easy to read. Please substitute all of figures to high resolution ones. Legends of some figures or table (Fig. 2 etc.) are required more detailed explanation.

Re: We have substituted all of figures with a higher quality resolution. Besides, we also added more detailed explanation in the legends of some figures or table.

Thanks for your valuable comments.

Reviewer 3 Report

The manuscript “Mapping invasion potential of the pest from Central Asia, Trypophloeus klimeschi (Coleoptera: Curculionidae: Scolytinae), in the shelter forests of Northwest China” presents species distribution models of the populus bark beetle Trypophloeus klimeschi under climate change scenarios in China. Generally, the work is well written, however, some changes I would like to recommend that would improve the work: First of all, I miss information about the native occurrence of the beetles and the inclusion of this native geographic occurrences in the modelling, because the bioclimatic conditions of a species' natural distribution may tell us more about its potential future distribution than does its non-native distribution alone. Thus, we can not be sure, if the current distribution in China does reflect the potential, suitable climatic niche.

Some minor comments:

Introduction

Line 58/59: “Recently, surveys indicated that the rate of trees with T. klimeschi was as high as 61.5%” The sentence is vague. Where exactly? In China? Everywhere where these beetles are distributed?

Material&Methods

Paragraph 2.2: You are talking about the relevant literature (line 103), “collected from the literature” (l 114), “from the literature” (l115, l123), “we consulted the literature” (l122). Please, cite the literature!

Results

The quality of the maps is weak.

Discussion

Line 337: …are likely to cause its development retardation.

Author Response

Dear Reviewer:

Thank you for your letter and for the reviewers’ comments concerning our manuscript entitled “Mapping invasion potential of the pest from Central Asia, Trypophloeus klimeschi (Coleoptera: Curculionidae: Scolytinae), in the shelter forests of Northwest China” (ID: insects-118240). Those comments are all valuable and very helpful for revising and improving our paper, as well as the important guiding significance to our researches. We have studied comments carefully and have made correction which we hope meet with approval. Revised portion are marked in red and added portion are marked in blue in the manuscript.

The main corrections in the manuscript and the responds to the reviewer’s comments are as flowing:

The manuscript “Mapping invasion potential of the pest from Central Asia, Trypophloeus klimeschi (Coleoptera: Curculionidae: Scolytinae), in the shelter forests of Northwest China” presents species distribution models of the populus bark beetle Trypophloeus klimeschi under climate change scenarios in China. Generally, the work is well written, however, some changes I would like to recommend that would improve the work: First of all, I miss information about the native occurrence of the beetles and the inclusion of this native geographic occurrences in the modelling, because the bioclimatic conditions of a species' natural distribution may tell us more about its potential future distribution than does its non-native distribution alone. Thus, we can not be sure, if the current distribution in China does reflect the potential, suitable climatic niche.

Re: We have made revised and adjust according to your comments. T. klimeschi is native to the Kyrgyz Republic, which borders Xinjiang Uygur Autonomous Region. Specifically, its geographical location is connected with Aksu and Kashgar area in Xinjiang and climate is the temperate continental climate. Similarly, the occurrence in northwest China also is the temperate continental climate with the Gobi Desert as the main landform. Therefore, the suitable climatic niche in northwest China provides the prerequisite for the spread of the beetle. However, there have been few reports of the beetle in Central Asia in nearly 90 years. Until 2003, the beetle was found in Aksu Prefecture, Xinjiang. Then, the beetle began to gradually attract the attention of related scholars because of its devastating damage to Populus alba var. pyramidalis Bunge. However, reports on this pest in China are also very limited. In the sampling, we also tried to go to Central Asia to carry out some investigation, but it did not achieve. We have identified the occurrence point in Kyrgyz Republic through consulting and other means. We are now determining whether the beetle is distributed in other states, such as Eastern Europe and North America. However, these records can reflect the real climate niche because the current occurrence records in China and Kyrgyz Republic located in the same climate zone. So they can be used to predict the suitable area of China. But the next work is to focus on the global forecast of the beetle, and then compare the forecast in China, which may be more able to reflect the spread trend of the beetle.

Some minor comments:

Introduction

Line 58/59: “Recently, surveys indicated that the rate of trees with T. klimeschi was as high as 61.5%” The sentence is vague. Where exactly? In China? Everywhere where these beetles are distributed?

Re: We have made revised according to your comments. Because the reference of the rate is not clear, and the sample site of the survey only occurs in Dunhuang. It's confusing, so we delete the sentence

Material & Methods

Paragraph 2.2: You are talking about the relevant literature (line 103), “collected from the literature” (l 114), “from the literature” (l 115, l 123), “we consulted the literature” (l 122). Please, cite the literature!

Re: We have made added relevant literature according to your comments.

Results

The quality of the maps is weak.

Re: We have substituted all of figures with a higher quality resolution.

Discussion

Line 337: …are likely to cause its development retardation.

Re: We have made revised according to your comments. “…are likely to affect its seasonality

Thanks for your valuable comments.

Round 2

Reviewer 1 Report

The manuscript of Ning and colleagues entitled “Mapping invasion potential of the pest from Central Asia, Trypophloeus klimeschi (Coleoptera: Curculionidae: Scolytinae), in the shelter forests of Northwest China” substantially improved in quality and readability after revision. I would like to congratulate to the authors for their detailed answers to my comments. In the revised version of the manuscript figures are not included, but the latter can be due to problems in manuscript submission. I suggest to accept the manuscript after minor corrections:

Line 44: Substitute “invades” with “attacks”

Line 49: Here and elsewhere, substitute “T. klimeschi” with “Trypophloeus klimeschi” when reported at the beginning of a sentence. Moreover, substitute “has” with “completes”

Lines 64-67: Correct the sentence as: “Future changes in temperatures might have significant effects on insect growth rate, thus influencing both biological processes (e.g. number of generations per year) and geographical distribution.”

Line 208: Change the sentence as “The model prediction showed a good performance as the AUC values from training and test datasets were 0.898 and 0.876, respectively”

Line 322: Change “exists” with “grows”.

Line 325: Change “limited by” with “limited to”.

Line 328: Substitute “where there is no water and” with “where environmental conditions”.

Line 329: Delete the sentence “As the herbivorous insect, T. klimeschi can’t continue to occur in those places where their host plants are not present.”

Line 333: Change “generally can reach - 20°C can’t guarantee” with “(-20°C) might not allow”.

Author Response

Dear Reviewer:

Thank you for your letter again and for the reviewers’ comments concerning our manuscript entitled “Mapping invasion potential of the pest from Central Asia, Trypophloeus klimeschi (Coleoptera: Curculionidae: Scolytinae), in the shelter forests of Northwest China” (ID: insects-118240). Those comments are all valuable and very helpful for revising and improving our paper, as well as the important guiding significance to our researches. We have studied comments carefully and have made correction which we hope meet with approval. Revised portion are marked in red in the manuscript.

The main corrections in the manuscript and the responds to the reviewer’s comments are as flowing:

The manuscript of Ning and colleagues entitled “Mapping invasion potential of the pest from Central Asia, Trypophloeus klimeschi (Coleoptera: Curculionidae: Scolytinae), in the shelter forests of Northwest China” substantially improved in quality and readability after revision. I would like to congratulate to the authors for their detailed answers to my comments. In the revised version of the manuscript figures are not included, but the latter can be due to problems in manuscript submission. I suggest to accept the manuscript after minor corrections:

Re: Thank you very much for your general comments. I will pay attention to the problems you mentioned in next work.

Line 44: Substitute “invades” with “attacks”

Re: I have substituted it in the manuscript the according to your comments.

Line 49: Here and elsewhere, substitute “T. klimeschi” with “Trypophloeus klimeschi” when reported at the beginning of a sentence. Moreover, substitute “has” with “completes”

Re: I have substituted them in the manuscript the according to your comments.

Lines 64-67: Correct the sentence as: “Future changes in temperatures might have significant effects on insect growth rate, thus influencing both biological processes (e.g. number of generations per year) and geographical distribution.”

Re: I have revised the sentence in manuscript the according to your comments.

Line 208: Change the sentence as “The model prediction showed a good performance as the AUC values from training and test datasets were 0.898 and 0.876, respectively”

Re: I have revised the sentence in manuscript the according to your comments.

Line 322: Change “exists” with “grows”.

Re: I have revised it in manuscript the according to your comments.

Line 325: Change “limited by” with “limited to”.

Re: I have revised it in manuscript the according to your comments.

Line 328: Substitute “where there is no water and” with “where environmental conditions”.

Re: I have substituted it in manuscript the according to your comments.

Line 329: Delete the sentence “As the herbivorous insect, T. klimeschi can’t continue to occur in those places where their host plants are not present.”

Re: I have deleted it in manuscript the according to your comments.

Line 333: Change “generally can reach - 20°C can’t guarantee” with “(-20°C) might not allow”.

Re: I have revised it in manuscript the according to your comments.

Thanks for your valuable comments.

Reviewer 3 Report

The manuscript has been significantly improved. I have no further comments.

Author Response

Dear Reviewer:

Thank you for your letter again and for the reviewers’ comments concerning our manuscript entitled “Mapping invasion potential of the pest from Central Asia, Trypophloeus klimeschi (Coleoptera: Curculionidae: Scolytinae), in the shelter forests of Northwest China” (ID: insects-118240).

The manuscript has been significantly improved. I have no further comments.

Thanks for your valuable comments.
